# Influence of Porosity on the Elastic Modulus of Ti-Zr-Ta-Nb Foams with a Low Nb Content

**Claudio Aguilar** [1],*, **Mariette Arancibia** [1], **Ismeli Alfonso** [2], **Mamie Sancy** [3], **Karem Tello** [1], **Vicente Salinas** [4] **and Fernando De Las Cuevas** [5]

[1] Departamento de Ingeniería Metalúrgica y Materiales, Universidad Técnica Federico Santa María, Av. España 1680, Valparaíso C.P. 2390123, Chile; Mariet-te.arancibia@gmail.com (M.A.); karem.tello@usm.cl (K.T.)

[2] Instituto de Investigaciones en Materiales, Unidad Morelia, Universidad Nacional Autónoma de México, Campus Morelia UNAM. Antigua Carretera a Pátzcuaro No. 8701, Col. Ex-Hacienda de San José de la Huerta, Morelia C.P. 58190, Michoacán, Mexico; ialfonso@iim.unam.mx

[3] Escuela de Construcción Civil, Pontificia Universidad Católica de Chile, Avenida Vicuña Mackenna 4860, Santiago C.P. 8320000, Chile; mamiesancy@gmail.com

[4] Núcleo de Matemáticas, Física y Estadística, Facultad de Estudios Interdisciplinarios, Universidad Mayor, Manuel Montt 318, Providencia C.P. 7500628, Chile; vicente.salinas@uchile.cl

[5] Departamento de Física, Universidad Pública de Navarra, Campús de Arrosadua, C.P. 31006 Pamplona, Spain; fernando.delascuevas@unavarra.es

\* Correspondence: claudio.aguilar@usm.cl; Tel.: +56-32-2654228

**Abstract:** The development of titanium foams with a low elastic modulus has increased their scientific and technological relevance due to the evident need to avoid stress shielding problems. In this work, we studied the synthesis and characterization of Ti-13Zr-13Ta-3Nb (wt.%) alloy foams which present high potential for biomedical applications. A Ti-Nb-Ta-Zr mixture was produced by mechanical alloying using a planetary mill. Ti alloy foams were obtained using NaCl as a space-holder (40, 50, and 60 v/v %) that was mixed with the metallic powders and compacted under 420 MPa stress. NaCl particles were removed from the green compacts by submerging samples in distilled water at 60 °C. The green compacts were sintered at 1300 °C for 3 h in Ar atmosphere. Powders and foams were characterized by SEM and optical microscopy. The results showed that Ti-based foams with a tailored heterogeneous pore distribution can be obtained using the space holder method. The elastic modulus ($E$) of foams was estimated and measured between 5 and 25 GPa using theoretical and finite element analysis (FEA) models which are close to the $E$ values measured experimentally. The results showed that foams with 50% and 60% porosity are potential bone replacement materials because their $E$ value is closer to the E value of human bone.

**Keywords:** metallic foam; titanium alloys; elastic modulus; biomaterials; powder metallurgy

## 1. Introduction

Pure titanium and its alloys have been widely employed in aerospace, automotive, and biomedical applications due to their excellent combination of corrosion resistance, biocompatibility, and high specific mechanic strength [1]. However, for biomedical applications, the elastic modulus mismatch between Ti implants (~110 GPa) and human bone (1–30 GPa) represents an important disadvantage which limits their use. The high elastic moduli of Ti affect the bones, producing an effect called stress-shielding, which results in osteoporosis and decreased bone strength. One of the first Ti-based alloys developed for biomedical applications was Ti-6Al-4V. However, this alloy presents two issues: (i) an elastic moduli value around 110 GPa [2]; and (ii) Al and V ions are released into the human body,

which can produce long-term health problems, such as Alzheimer's disease [3], dermatitis, neuropathy, and ostemomalacia [4]. For this reason, V- and Al-free Ti-based alloys with alloying elements such as Nb, Zr, Sn, or Ta were explored for implant applications [2]. Among these already developed new titanium alloys, it is possible to find Ti-30Nb-13Ta-4.6Zr, Ti-35Nb-5Ta-7Zr, Ti-13Nb-14Zr [5], Ti-13Nb-13Zr, Ti-35Nb-7Zr-5Ta, Ti-Mo, Ti-29-Nb-13Ta-4.6Zr [4], and Ti-34Nb-29Ta-6Mn [6]. Nb, Zr, and Ta are used in these Ti-based alloys for two main reasons: (i) they exhibit a small biological impact [7]; and (ii) they stabilize the β-phase of Ti, which has a smaller elastic moduli than the α-phase [8]. Among the disadvantages that Ti-Nb-Ta-Zr alloys have are: (i) the high economic cost of Nb, Ta, and Zr elements; (ii) the imposition to keep the Ta content below 13% in order to maintain a low elastic moduli [9]; (iii) high amounts of alloying elements may lead to the precipitation of undesirable phases, such as omega phase (ω), which increases the elastic modulus of the alloy (ω) is a metastable phase with space group P6/mmm); and (iv) the lowest elastic modulus measured for Ti-Nb-Ta-Zr alloys is around 40-55 GPa [10], which is about half of the elastic modulus of Ti-6Al-4V alloys [5], but still higher than the cortical and cancellous bones.

In response to the limitation exhibited by Ti alloys, Ti-based metallic foams have risen as an attractive solution due to the possibility of controlling their elastic modulus [2] and decrease their stress-shielding [11]. Metallic foams can be defined as a mix of bulk walls with embedded pores that exhibit mixed properties in between metals and foams. For this reason, metallic foams can be classified as light-weight materials. Mechanical properties of foams can be tuned by controlling their porosity (pore size, pore distribution, pore shape, and pore quantity). The porosities are classified into two main categories: closed and open pores. In biomaterials, foams with open porosity are preferred because they promote osteointegration and cell adhesion [12]. Metallic foams with open pores can be obtained from solid, liquid, or gaseous phases [13] and the selection of the fabrication method depends on the type, distribution, and size of pores [14]. Based on the design of the foams, their elastic modulus values (*E*) can be predicted by different models proposed in the literature (see Table 1, which presents a summary of these various models). Synthesis of Ti-based foams via powder metallurgy became an attractive alternative. Powder metallurgy uses basically two approaches: (i) sintering of metal powders and/or (ii) addition of space holders. In the case of the space holder strategy, the size and distribution of pores and type of porosity (open and close pores) can be controlled by changing the shape, size, and packing of the space-holder; the sintering temperature; or the time and temperature of dissolution (if the space holder is salt) [15]. Some space holders commonly used to synthetize metallic foams are: ammonium hydrogen carbonate (NH5CO3) (used in open pore titanium foams [16], titanium foams with 60% porosity [17], porous NiTi shape memory alloys [18] and titanium foams with porosity between 40 to 70% [19]), carbamide (aluminum foams [20], titanium foams with porosities between 55 to 75% [21]), sodium chloride (used in aluminum foams [22],titanium foams with NaCl between 40 to 70% in fraction volume [15], titanium foams with open cell and NaCl volume fraction between 30 to 70% [23]), magnesium (used in Ti-6Al-4V foams with porosities between ~43 to 64% [24]), potassium carbonate (K2CO3) (used in copper foams with porosity in the range 50–85% [25]), potassium sorbate (used in Ti-6Al-4V foams [24]), polymethyl methacrylate (used in titanium foams with 50% porosity [26]), and tapioca starch (used in titanium foams with porosities between 64 to 79% [27]).

The space holder must be selected based on the following criteria: (i) size and shape, (ii) cost, (iii) reactivity with Ti, (iv) presence of residue, (v) easy processability, and (vi) non-toxicity. Carbamide, for instance, has been used to synthetize Ti foams [28] due to its low cost, extensive availability as spherical granules, easy leachability in water, and lack of reactivity with Ti [29]. However, the foams obtained have closed pores as a consequence of the characteristic shape of carbamide granules. Mg can also be used as a space holder due to its relatively low melting/dissociation temperatures and its ability to be removed after thermal treatment. Esen and Bor [24] studied the synthesis of Ti-6Al-4V foams using Mg as a space holder in different amounts. They found that the total porosity of foams was composed of macro and micro pores, which formed as a result of the evaporation of magnesium and

incomplete sintering of spherical powders present in cell walls and edges, respectively. Nevertheless, Mg is prone to oxidation, which causes degradation of the mechanical properties of the material and leaves residues that cannot be completely removed. Anther space holder used is NH5CO3 because it decomposes at relatively low temperatures (200 °C). In addition, it exhibits a low toxicity and non-reactivity with Ti. Wang et al. [30] studied the effect of $NH_5CO_3$ addition on Ti-10 wt% Mg foams. They found that the porosity increased with the amount of $NH_5CO_3$, while open porosity increased with particle size of the space holder. Some works have reported that NaCl particles are excellent space holders for making highly porous open cell foams [31] because they have the ability to be completely removed through the dissolution process [32]. In addition, NaCl is not expected to react with Ti, exhibits a low cost, presents fast dissolution in water, reduces the etching of metal during dissolution, and has a low toxicity.

The goal of this work is to study the synthesis and characterization of Ti-based alloy foams with a low Nb content. In particular, Ti-13Zr-13Ta-3Nb (wt.%) alloy foam was chosen because the effect of porosity on its elastic modulus has not been reported. NaCl was used as the space holder in different concentrations in order to obtain a varied range of porosity. In addition, the topology of the foams was modeled using a randomly arranged porous network and the compressive Young´s moduli of the foams were estimated using finite element analysis (FEA). The models were compared and validated with the experimental results.

## 2. Materials and Methods

### 2.1. Finite Element Model

FEA was used to predict the elastic modulus before manufacturing the alloy. Therefore, it is a key tool for estimating the mechanical behavior of the foams as a function of their porosities. The FEA models used considered random coordinates because they grasp the real nature of the pore arrangement of real foams. The coordinates were generated using GraphPad Software (https://www.graphpad.com/, Prism 8, GraphPad Software Inc., San Diego, CA, USA), which allowed the random generation of $x$, $y$, and $z$ coordinates. These coordinates were located inside a confined cylinder that represents the foam and each $(x, y, z)$ coordinate coincides with each pore. Subsequently, the system foam/pores were modelled using ANSYS 17.2 FEA software (Canonsburg, PA, USA) in which spheres of a 0.3 mm diameter were in the coordinate system described above with each center of the spheres being related 1:1 with each coordinate. On the other hand, the metal was represented as a cylinder of 2 mm in diameter and 4 mm in height. Finally, the FEA model of the foam with a given porosity was obtained by subtracting the volume of the resulting spheres from the volume of the cylinder. The previous procedure was carried out for each intended porosity, namely 40, 50, and 60%. A diameter/height ratio of 0.5 was used in order to replicate the cylindrical samples obtained in the present work. Figure 1 shows the random distribution of spheres used in the model. These reduced models were selected in order to optimize computational requirements and meet the condition that representative volume elements (RVE) must be at least ~2.5–3.5 times the maximum size of the certain feature that represents the material (an individual pore in our case) [33]. The size of the pores was modeled to reproduce the size of the space holder.

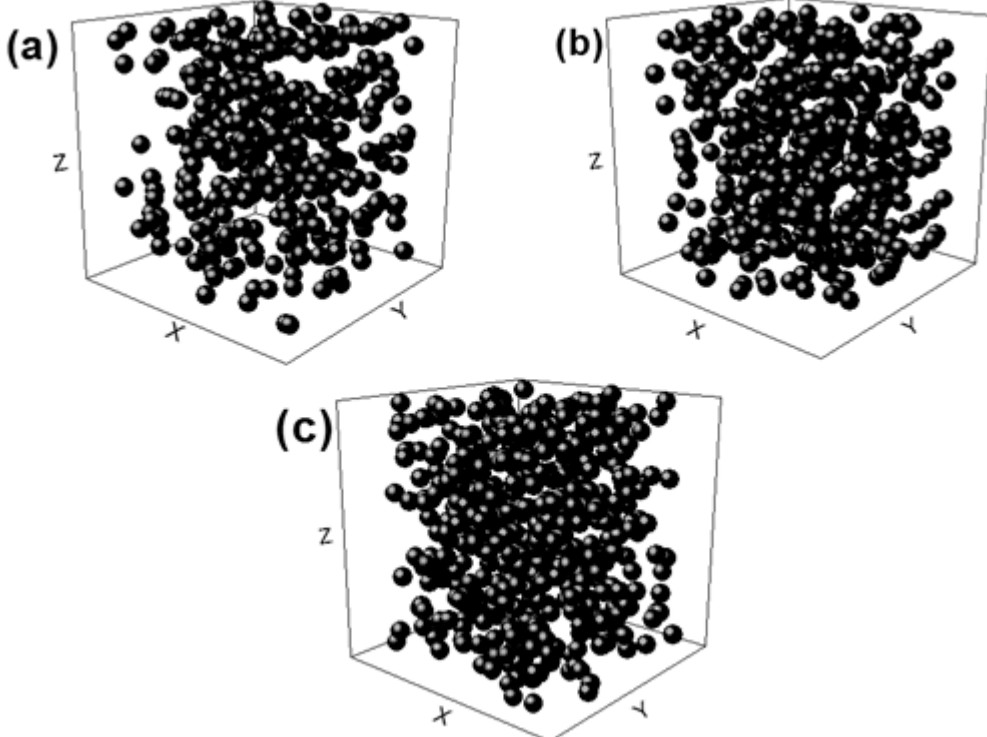

**Figure 1.** Sphere distributions used to generate the porosity of the foam. (**a**) 40%, (**b**) 50%, and (**c**) 60% porosity, respectively.

Young's moduli of the metallic foams with different porosities were estimated by applying equivalent compressive stresses of 20 MPa on the upper end nodes of the cylindrical specimens. The 187 3-D 10-node tetrahedral structural solid element was employed for meshing. Mesh convergence analysis was carried out by incrementally increasing the number of elements and verifying the estimations, to ensure convergence of the numerical solution. The coupled-node boundary condition (keeping the nodes in the same plane) was used for the upper face of the cylinder. This condition is applied since the presence of pores causes un-even surfaces and makes the deformation measurement hard to define. The nodes of the bottom were constrained for *z* displacement, while the movement of the surfaces of the internal hole was radially fixed. Young's modulus was obtained from the response of the compression test along the *z*-axis using the resulting displacements. Young's modulus (115 GPa) and Poisson Ratio (0.38) used for simulations were obtained from a specimen sintered without space-holder particles (SHP), resulting in the absence of induced porosity.

*2.2. Synthesis and Characterization of Foams*

Synthesis of Ti foams was performed using 100 mesh (<147 μm) Ti powders of grade IV, 325 mesh (<43 μm) Ta powders of 99,9% purity, 325 mesh (<43 μm) Nb powders of 99,9% purity, and 325 mesh (<43 μm) Zr powders of 99,2% purity. Powder blends of Ti-13Zr-13Ta-3Nb (wt.%) alloys were prepared, placed in a 250 mL hardened steel vial, and milled for 12 h in argon atmosphere using a planetary mill (Retsch, PM400, Retsch GmbH, Haan, Germany). 1 wt.% of stearic acid was used as the process control agent. Different diameters of hardened steel balls (19.5, 15, 12, 11, and 8.0 mm) were used during the milling process with a constant ball/powder ratio of 10:1. Milled powders were mixed in proportions of 40, 50, and 60 v/v% with NaCl particles, used as a space holder. Cylindrical samples of a 13 mm diameter and 25 mm length were made from the powder mixture (milled powder with NaCl particles) by applying three different levels of uniaxial compaction stress, namely 317, 420, and 605 MPa, using a Zwick Roell (model Z030, Ulm, Germany) machine standard. The powder

mixture was fed into a steel die and then compacted. Zinc stearate was used as a lubricant between the punch and die. NaCl particles were removed using distilled water via the dissolution technique. This technique consisted of immersing the samples in distilled water at 60 °C, five times (in cycles of 4 h each). Distilled water was changed after each dissolution cycle. In addition, after each immersion cycle, the samples were dried at 110 °C for 30 min, and later, they were weighted. The sintering was performed in a furnace with an ultra-pure Ar atmosphere (<3 ppm $O_2$, <5 ppm $N_2$, <0.5 ppm other gases) at 1300 °C for 3 h.

The morphology of the powders and foams was studied using scanning electron microscopy (SEM, Carl-Zeiss, model EVO MA 10, Oberkochen, Germany). Transmission electron microscopy (TEM) characterization was carried out using FEI Tecnai, model F-20 FE-TEM equipment. Samples for TEM were prepared by suspending powders in isopropyl alcohol and placing the suspension on a Cu grid. The porosity of the foam was measured by image analysis software from images obtained with an optical microscope Leitz, model Metallux (Leitz GmbH & Co. KG, Oberkochen, Germany) coupled with a Leica MC170 HD camera (Wetzlar, Germany). For metallographic characterization, the samples were grinded using silicon carbide paper following the sequence 320, 400, 600, and 1000 grit. Later, the samples were polished following the sequence: (i) 9 µm diamond particles+ water and (ii) magnesium oxide + hydrogen peroxide (30%) using billiard cloths.

The elastic modulus was measured using ultrasonic transducers (Olympus, model V157, Tokyo, Japan); one as an emitter and another as a receiver located on opposite (and parallel) faces of the sample to be characterized. The emitter was excited with a pulse train of three to five cycles at a fixed frequency of 480 kHz, which generated mechanical (elastic) waves in the material (generator Agilent 33250A (Agilent Technologies, Santa Clara, CA, USA), amplifier NF model HSA4011 (NF, Japan) and oscilloscope Tektronix, model TDS2021b (Tektronix, Beaverton, OR, USA). The measurement and calculation protocol was based on the ASTM D2845-08 standard, which considers the following equations to correlate acoustic measurements with the mechanical properties of the measured samples:

$$v = \frac{1 - 2\left(\frac{v_T}{v_L}\right)^2}{2 - 2\left(\frac{v_T}{v_L}\right)^2} \tag{1}$$

$$E = 2\rho v_T^2 (1 - v) \tag{2}$$

where $v_T$ is the propagation velocity of the transverse wave, $v_L$ is the longitudinal wave velocity, $\rho$ is the density of the material, $E$ is the Young's modulus, and $v$ is the Poisson's ratio. The density measurement was performed using a 20 mL pycnometer and a balance with 0.001 g sensitivity, based on the ASTM D792-08 standard. Additionally, compressive tests were carried out to determine the yield strength of the foams using a Zwick-Roell machine, model Z030 at a loading rate of 0.02 mm/min, according to the BS ISO 14317:2015 standard.

## 3. Results and Discussion

### 3.1. Characterization of Starting Powders

Figure 2a shows the particle size distribution of NaCl used as a space holder. The NaCl particle size ranges from 150 to 350 mm, which is an acceptable pore size for metallic foams used in biomedical applications. Such macro pores have been reported to induce cell adhesion, the growth of new bone tissues, and vascularization [12]. Figure 2b shows the relative mass loss (Ti + NaCl) of compacted samples as a function of three variables; immersion time (cycle of 4 h), compaction stress, and the amount of NaCl (40, 50 and 60v/v%). Relative mass decreased when the immersion time increased for all amounts of NaCl and compaction strengths. Samples with the lowest compaction strength needed a shorter dissolution time, independent of the amount of NaCl and immersion time, because a low compaction strength produces a greater porosity in green samples. Another effect observed is

the increase of dissolution rate when increasing the amount of NaCl for processes shorter than 8 h (2 cycles). The optimal extension of immersion time was found at five cycles (20 h) when relative mass changes were small. Extension of the immersion time (after six cycles) led to the loss of sample structural integrity. Four compaction strengths were tested and the optimal compaction strength was found at 420 MPa, where samples kept their structural integrity. For a higher compaction strength, the time to dissolve NaCl particles was higher. For lower strength values, samples lost their structural integrity. These results are in good agreement with Torres et al. [32], who studied NaCl dissolution in commercial pure Ti foams and found that the NaCl dissolution rate was influenced by the immersion time per cycle and was less sensitive to the compaction strength for longer immersion times.

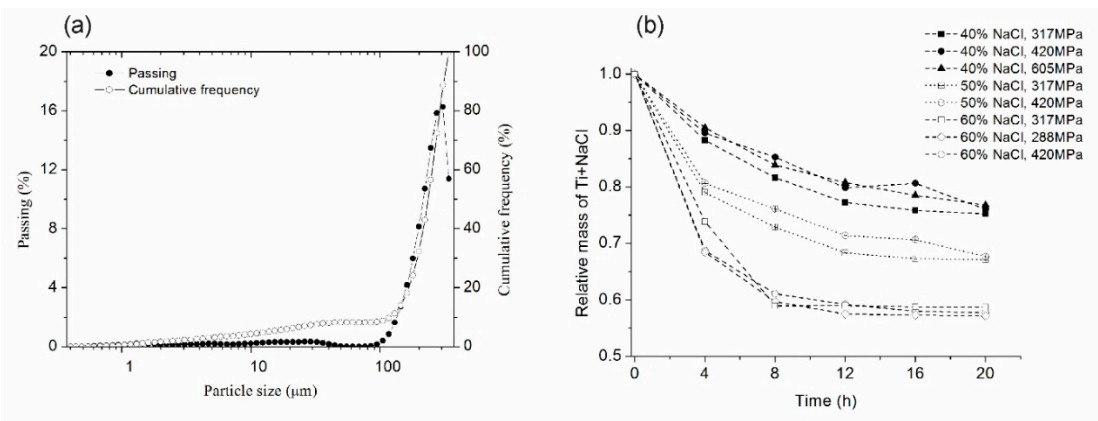

**Figure 2.** (**a**) Particle size distribution of NaCl used as a space holder and (**b**) relative mass loss of Ti+NaCl as a function of dissolution cycles in distilled water at 60 °C.

Figure 3 shows SEM images of as-received elemental Ti, Nb, Ta, and Zr powders, together with NaCl powder used as a space holder. Ta and Ti exhibited the largest particle size (200 μm and 150 μm, respectively) when compared with the other elements. In terms of morphology, Ti powder showed an irregular shape, whereas Ta powder exhibited a sponge-like aspect. Nb and Zr powders had a particle size smaller than 45 μm and showed an irregular morphology. While, NaCl particles were smaller than 350 μm and exhibited a cubical morphology. The starting particle size of metallic powders was larger than that for milled metallic particles (Figure 4). In contrast, NaCl particles were larger than milled particles, which is an important criterion to form metallic foams.

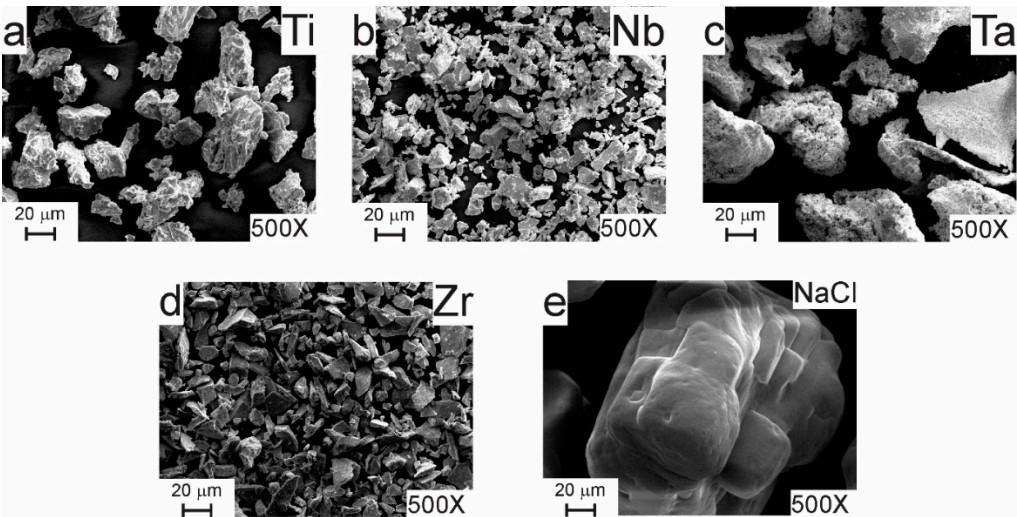

**Figure 3.** SEM images of as-received (**a**) Ti, (**b**) Nb, (**c**) Ta, (**d**) Zr, and (**e**) NaCl powders.

### 3.2. Characterization of Milling Powders

Figure 4a–c shows the morphology of Ti-13Zr-13Ta-3Nb powders after 12 h milling. Figure 4b,c shows the agglomeration of particles with an irregular shape and particle size ranging from 10 to 40 μm. In addition, the edges of the particles were smooth, which indicates that cold welding was more relevant than fracture during mechanical alloying (MA). In addition, particles are repeatedly flattened, cold welded, fractured, and rewelded during the milling process. Kinetics and predominance of cold welding or fracturing at any stage mostly depends on the characteristic deformation of the starting powders [34]. Fe contamination was detected through EDS analysis (Figure 4d). This is expected when steel vials are used in MA.

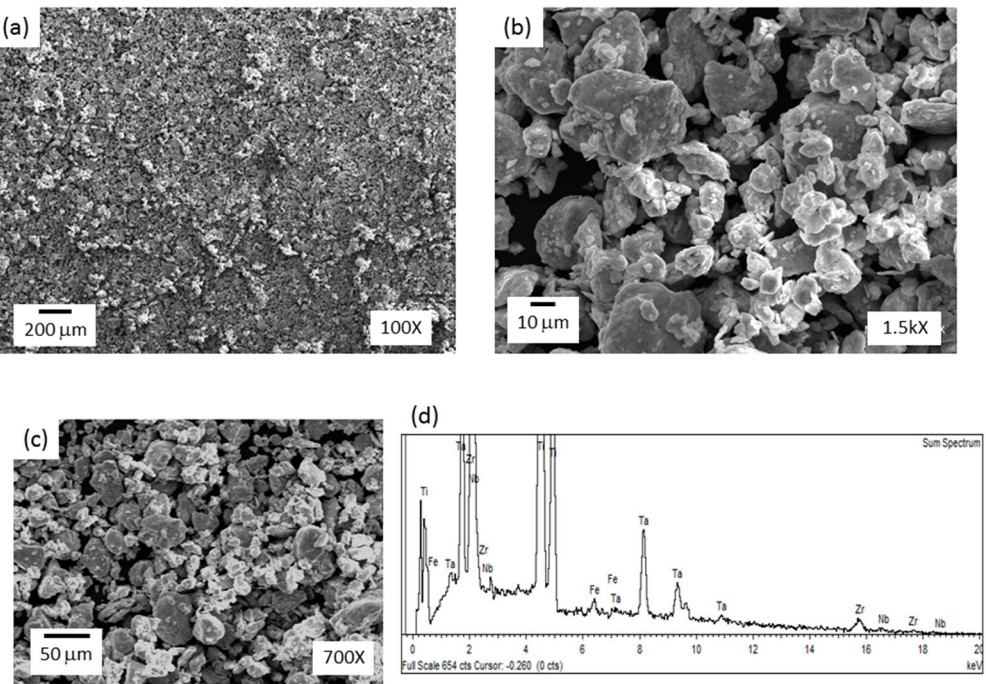

**Figure 4.** SEM images of milled Ti-13Zr-13Ta-3Nb powders and corresponding EDS analysis.

### 3.3. Characterization of Foams

Figure 5 shows detailed SEM images of synthesized foams with different magnifications (all images are over radial cross sections). The images show the presence of both a macro pore smaller than 600 μm (Figure 5a–c) and micro pore smaller than 20μm (Figure 5d–f). The macro pores were produced by removing NaCl particles through the dissolution technique and the micro pores were produced as a consequence of the sintering process. Macro pores were not uniformly distributed and their size was larger than individual NaCl particles, indicating that NaCl agglomerates during the mixing process. The microstructure of foams presents the following characteristics: (i) micro and macropores have rounded corners, which reduce the stress concentration; (ii) pores exhibit a rough inner surface, which is relevant for bone growth [35], (iii) macropores display an equiaxial shape, which can be interpreted as a disadvantage when compared with pores in human bones that reveal an elongated morphology [36]; and (iv) micropores connection leads to macropores interconnectivity (Figure 5d–f). The interconnectivity values were measured by optical image analysis using the Cpore parameter (this value when interconnectivity does not exit is zero and 1 when the macropores are connected). The values measured were between 0.3 to 0.4 for all foams. This interconnectivity is an important requirement for porous implants because it promotes the nutrition and transport of body fluid.

The pores' main characteristics (size distribution, morphology, interconnectivity, inner surface roughness, among others) can be controlled by modifying the shape, packing, and size of the space

holder; type and amount of metallic powders; dissolution process; and heat treatment. Similar results for different foam systems have been reported. Jha et al. [31] synthetized foams of pure Ti using NaCl as a space holder and reported a macropore size of around 250 μm, whereas Ye and Dunand [23] obtained Ti-6Al-4V foams using NaCl with a macro and micropore size around 500 and 50 μm, respectively. Both research groups reported the synthesis of foams by means of hot pressing, which generates pores of an elongated shape due to the stress applied during compaction.

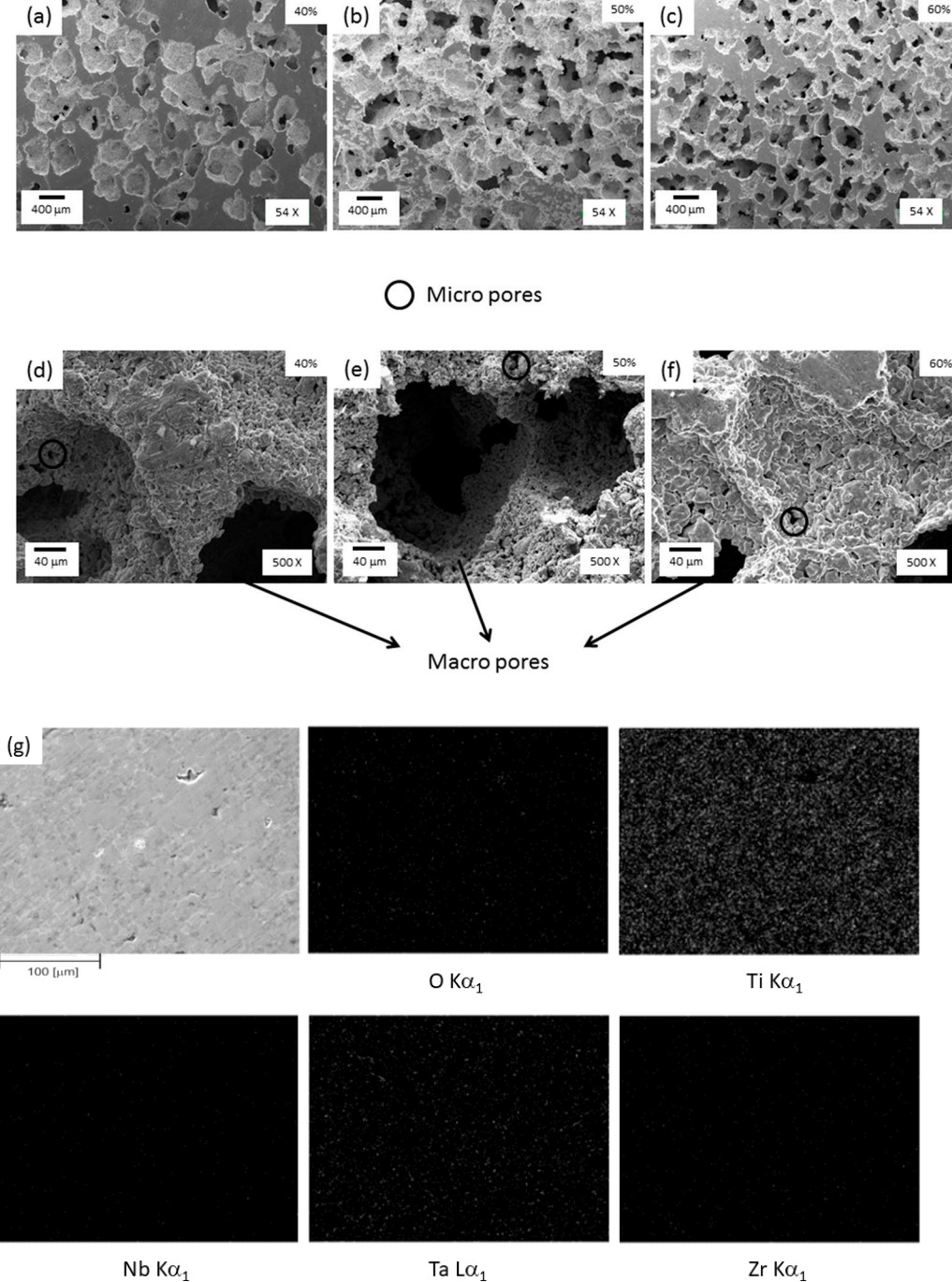

**Figure 5.** SEM images of foams with (**a,d**) 40, (**b,e**) 50, and (**c,f**) 60% porosity with different magnifications and (**g**) EDS analysis for the 60% porosity foam sample.

The spatial distribution of foam composition was obtained using the EDS/SEM mapping method. Figure 5g revealed that Ti, Nb, Ta, and Zr are spread homogeneously within the sample with 60%

porosity. The same compositional trend was observed in the foams with 40 and 50% porosity. Also, the presence of oxygen was detected within foams, probably connected to the high chemical affinity with oxygen of all elements present in foams, which led to oxide formation due to their low equilibrium partial pressure of oxygen.

Figure 6 shows optical microscopy images of foams where it is possible to observe the typical characteristics of foams fabricated by the space holder method [13], such as: (i) presence of zones with a low density of macro pores; (ii) macropores larger than NaCl particles, indicating that agglomeration was produced; and (iii) irregular thickness of cell edges produced by irregular macropores. Cell wall thickness, cell size, and the distribution of pores change as a function of the Ti/NaCl ratio, similarly to results reported by Jha et al. [31], who obtained macropores smaller than 600 μm. These results are in good agreement with previous reports for TiZr foams synthesized by the space holder method, where macropores around 200–500 μm were observed [12]. In the case of Ti-16Sn-4Nb foams, Nouri et al. [36] also reported the presence of rounded macro pores of around 350 μm. Macro pores were interconnected in all samples, whereas micro pores were distributed at cell edges of macro pores. This kind of bimodal porous microstructure is desired for vascularization and osteointegration processes.

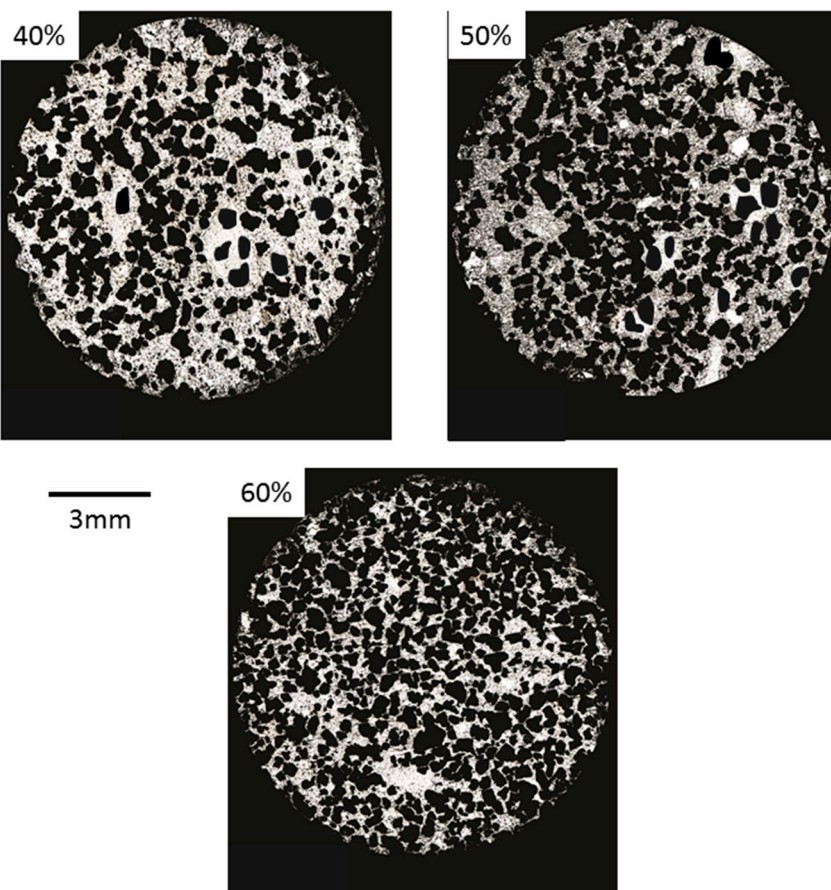

**Figure 6.** Optical images of 40, 50, and 60% porosity foams.

Figure 7 shows the results of pore morphology obtained from optical image analysis, where *P(IA)* is the estimated total porosity; *Deq* is the equivalent diameter (average diameter measured from the pore centre); *Dp* is the density of the pore or pore quantity measured by unit of area; *R* is the roundness; *Ff* is the shape factor, which was determined using the expression $Ff = 4\pi A/(PE)2$ (A is the pore area and *PE* is the experimental perimeter of the pore); λ is the mean free path between the pores (measure of the mean size of titanium matrix); and *Cpore* is the pore contiguity (a measurement of pore interconnectivity) [37]. The highest frequency value of *Deq* was 10 μm. The pores exhibited

three characteristic sizes regions, which are related to sample processing: (i) between 350 to 600 μm, macropores formed due to the agglomeration of NaCl particles; (ii) between 100 to 350 μm, macropores formed as a consequence of NaCl particles dissolution; and (iii) micropores smaller than 20 μm formed during the sintering process. It was found that the majority of the pores have sizes smaller than 200 μm, which explains the maximal value of 300 μm in *Deq* (abscissa) in Figure 7. *Ff* (shape factor) and *R* (convexity) values are close to 1, which is in agreement with the semi-equiaxial morphology showed by pores. These results can be interpreted in terms of the cubic morphology of NaCl particles, which affects the way that particles fracture during compaction and, later, are re-arranged after compaction. The free-mean path or distance between pores (λ) is around 10 μm, whereas *Cpore* values are between 0.3 and 0.4, which indicates interconnectivity within foams. This parameter value is 0 for null interconnectivity and 1 for maximum. Generally, the *Cpore* value matches the corresponding measurement obtained by Archimedes' method, making *Cpore* a reasonably good parameter for estimating pore interconnectivity. This parameter is also related to the distribution of equivalent diameter; if *Cpore* increases (*Cpore* →1 for highest interconnectivity), the pore size distribution becomes wider [38]. The fact that *Ff*, λ, *R*, and *Cpore* values are constant for the three foams can be understood in terms of the synthesis conditions, which for these samples, remained unchanged. *Ff*, λ, *R*, and *Cpore* values can be controlled by changing the shape, packing, and size of the space holder; amount of powders; dissolution process; and heat treatment. Finally, the measured porosity value was higher than expected (for the amount of space holder used) because the total porosity was determined by the contribution of the initial NaCl space holder particles and the porosity formed between particles due to the sintering process.

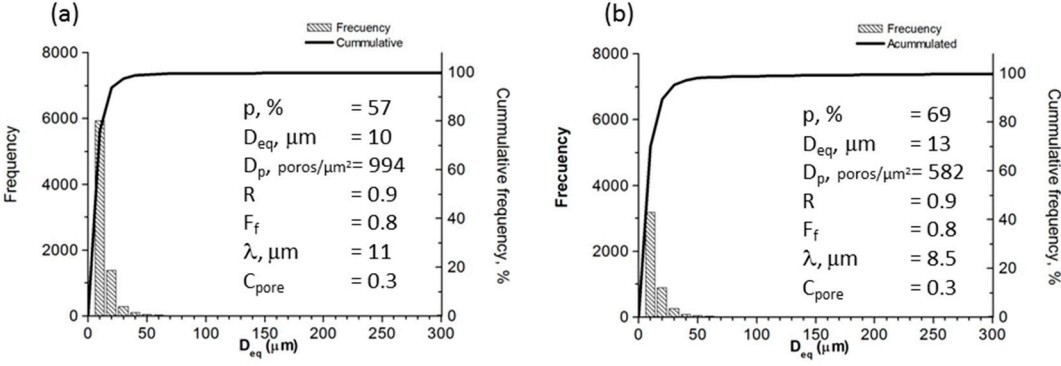

**Figure 7.** Results of optical image analysis of foams designed with (**a**) 50% and (**b**) 60 % porosity.

*3.4. Modeling and Measuring the Elastic Modulus of Foams*

　　Diverse theoretical models, such as Knused-Spriggs [39,40], Gibson and Ashby [41], Phani and Niyogi [42], Pabst and Gregorová [43], and Nielsen [44] can be used to predict the elastic modulus of metallic foams. The equations and constant values related to each model are listed in Table 1. Interpretation of each constant can be obtained from the corresponding references. These models were used in this study because they have been reported to provide *E* values close to the corresponding experimental measurements done in pure Ti [38] and Ti-6Al-4V [45]. Estimated *E* values were compared to the ones obtained using FEA. The accuracy of FEA estimations, compared to the experimental values, is connected to the proximity of the models to the real foam topologies. Figure 8 shows that the morphologies of real (upper images) and modeled (lower images) pore networks are very similar for each porosity percentage, suggesting the effectiveness of the method used for modeling foams. The presence of abundant pores interconnection was modeled as well, in agreement with features observed in SEM images. Interconnection is a significant factor for improving the prediction capabilities of FEA. Low interconnection in FEA models with a regular distribution is one of the most important causes of over predicting the Young's moduli.

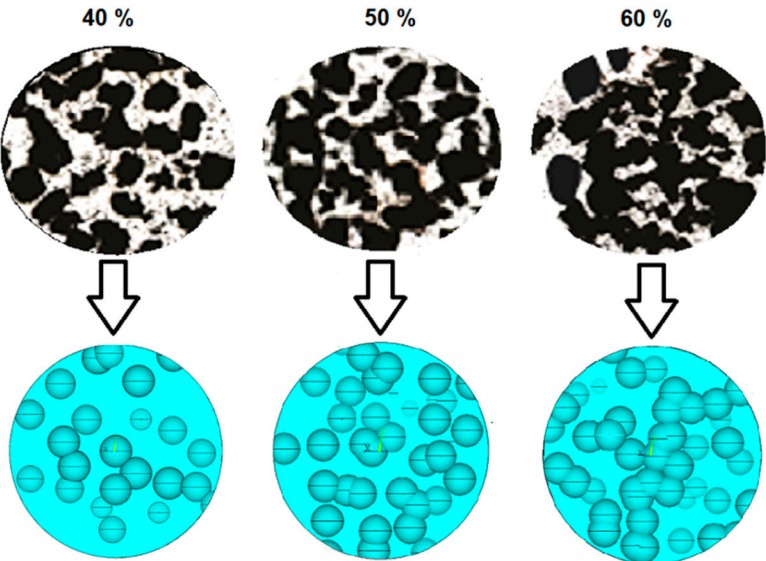

**Figure 8.** Comparison between cross section micrographs (up) and FEA modeled foams (bottom), for different foam porosities.

Figure 9 shows the *E* values estimated using theoretical models (Table 1), the *E* values obtained using FEA, and those experimentally measured in this work. The constant values, taken from the work of Zhu et al. [45], are listed in Table 1. In that report, E values of metallic foams with various porosities were determined using an electromagnetic acoustic resonance experiment. Such model and experimental data showed that the elastic modulus decreases when porosity increases. In our work, the Knudsen-Spriggs and Gibson-Ashby models provided *E* values close to experimental data (between 3 to 10 GPa). The Pabst-Gregorova, Nielsen, and Phani-Niyogi models exhibit higher *E* values compared with experimental data. All foams showed elastic modulus values in the range of cortical bones [46], which is a clear indication of their potential for biomaterial applications. It can be clearly observed that the FEA model estimations are very close to the experimental results, obtaining low errors compared to other used estimations. This result shows that FEA is an important tool for the study of foams, predicting their compressive behavior in an accurate way. The selection of the foam topology has been demonstrated to be an essential variable for obtaining a correct estimation without overpredicting foam strength, and many other models did not take into account pores' interconnection.

**Table 1.** Values of the constants used in different models reported in the literature to estimate the value of *E*.

| Model | Equation | Constants | Density (g/cm³) | E (GPa) at 0% porosity | p, % |
|---|---|---|---|---|---|
| Knudsen-Spriggs | $E = E_0 e^{-bp}$ | $b = 6.4$ | | | |
| Gibson-Ashby | $E = \alpha E_0 \left(\frac{\rho}{\rho_0}\right)^n$ | | 3.6 (40%p) 2.9 (50%p) 2.5 (60%p) | 115 | 40, 50, 60 |
| Phani-Niyogi | $E = E_0 \left(1 - \frac{p}{p_c}\right)^m$ | $m = 2.2$ $p_c = 70\%$ | | | |
| Pabst-Gregorová | $E = E_0 (1 - a\,p)\left(1 - \frac{p}{p_c}\right)$ | $a = 1$ $p_c = 70\%$ | | | |
| Nielsen | $E = E_0 \left[\frac{(1-p/100)^2}{1+\left(1/F_f - 1\right)p/100}\right]$ | $F_f = 0.79$ (40%p) $F_f = 0.84$ (50%p) $F_f = 0.80$ (60%p) | | | |

Where *p* is porosity; $\rho$ and $\rho_0$ are the density of the foam and dense matrix material, respectively; *c* and *n* are constants depending on the porous structure; $p_c$ is the critical porosity at which *E* = 0 (i.e.,

when the material loose its integrity); *a* is the packing geometry factor; and *Ff* is the pore shape factor ($Ff = 4pA/PE2$, *A* is the pore area and *PE* is the experimental perimeter of the pore).

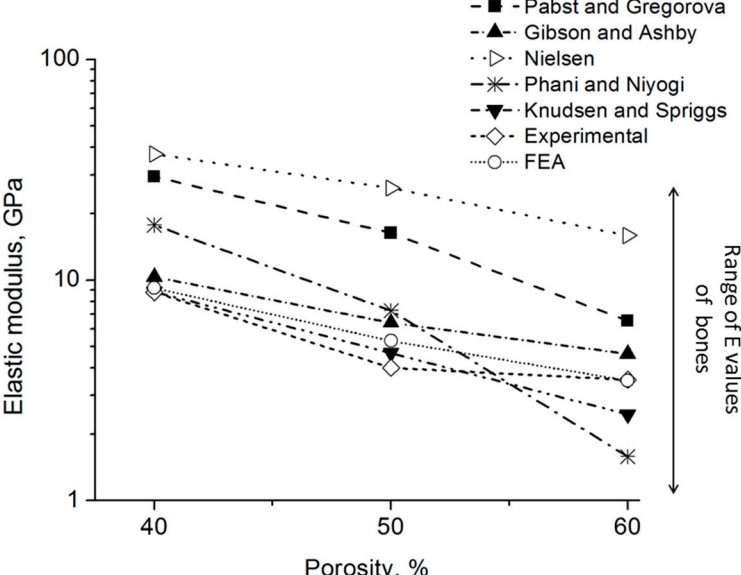

**Figure 9.** Estimation of E value using different models and experimental data.

The compressive strength varies as a function of porosity level. The samples with 40% porosity exhibit a higher compressive strength than the samples with 50% and 60% porosity. The compressive strength values measured were 164, 115, and 72 MPa for 40, 50, and 60% porosity, respectively. In addition, the samples did not exhibit a clear plastic yield stage (plateau stage) because the pore walls deformed plastically. The foams have pre-existing micro flaws in the pore walls, which decrease their strength. Therefore, a higher porosity results in a larger amount of flaws and, consequently, a lower strength. These results showed that the foams with 40% and 50% porosity have higher compressive values than the human bone (120 MPa), which means that can be used as biomaterial.

## 4. Conclusions

Ti-13Zr-13Ta-3Nb alloy foams were synthetized using NaCl as a space holder. The foams showed a non-homogeneous distribution of macro pores and micro pores with an irregular shape. Macro pores, produced by the introduction of the space holder, were smaller than 600 μm, while micro pores formed during the sintering process were smaller than 20 μm. Morphological characteristics of the macro and micro pores obtained are compatible with cell adhesion and bone growth.

The elastic moduli were calculated using various models and the results agree with previous reported values for other metallic foams. The calculated E values ranged from 8 to 40 GPa, 4 to 30 GPa, and 2 to 20 GPa for 40, 50, and 60% porosity, respectively; while the experimental values were 9, 5, and 4.5 GPa for the same porosity levels. Samples with 40% and 50% porosity exhibited compressive strength values above 120 MPa, which is the strength level of human bones. This suggests that Ti-13Zr-13Ta-3Nb alloy foams, synthesized via the space holder method, with 50% and 60% porosity, have a great potential for biomaterial applications, considering that the elastic modulus and compressive strength of foams were close to the elastic modulus of cortical bones.

The use of random FEA enables us to model a realistic topology of the foams, which is an essential variable for obtaining mechanical property estimations closer to the experimental measurements.

**Author Contributions:** Investigation, C.A., M.S., V.S., F.D.L.C., and M.A.; Methodology, M.A., Software, and I.A.; writing—review and editing, C.A. and K.T.

**Funding:** The authors would like to acknowledge financial support from FONDECYT Project N° 1161444 and USM PI-INN-18-02.

**Conflicts of Interest:** The authors declare no conflict of interest.

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
