# Peer review of "Influence of Porosity on the Elastic Modulus of Ti-Zr-Ta-Nb Foams with a Low Nb Content"

_metals, doi:10.3390/met9020176_

Reviewer 1 Report

The paper deals with a method to prepare a titanium foam material for biomedical applications.

The paper is well written and clear, well balanced, organized and complete.

The contents are interesting and all the steps are clearly explained.

Maybe, if only one point has to be raised, the finite elements model in §2.2 does not seem in the right place there: it is not clear whether the model was used to design the method or to forecast the material properties or what. 

As a final suggestion, maybe other material properties should be examined: for example, what can be said about the strength of this foam? How much porosity reduce the compression strength? Is it still compatible with bone strength? 

Author Response

1) Maybe, if only one point has to be raised, the finite elements model in §2.2 does not seem in the right place there: it is not clear whether the model was used to design the method or to forecast the material properties or what.

Response:  You are right. This section was included in order to forecast the material properties. Then, it was re-located as 2.1, while 2.1 Synthesis and characterization of foams now is 2.2 in the Materials and Methods section.

 2) As a final suggestion, maybe other material properties should be examined: for example, what can be said about the strength of this foam? How much porosity reduce the compression strength? Is it still compatible with bone strength?

Response: The other mechanical property is the compressive strength. The values were added and discussed in the manuscript.

Reviewer 2 Report

Porous titanium alloys are widely used for various medical applications. However, the known porous materials based on titanium in some cases do not meet the requirements of a particular use; The level of strength and elastic properties of porous elements under high stress conditions often does not meet the specified criteria of operation. When increasing the volume of bone tissue, it is necessary to ensure that the behavior of the material is similar to living tissue: showed sufficient elasticity, permeability, biochemical and biomechanical compatibility with the environment of the body. In this connection, the submitted manuscript, aimed at the synthesis and characterization of Ti-based alloy foams with small Nb content, is certainly topical. The data obtained by the authors are of interest to researchers engaged in the development of new porous materials for medical purposes. The illustrations and tables given in the manuscript fully disclose and confirm the results of the research. It is worth of publication, but some revisions are suggested. More specifically: 

1) lines 49, 50: Symbols, denoting the modification of the titanium phase, are missed;

2) line 53: What is the omega phase, what is its composition? 

3) line 113: What is -100 mesh powders, -325 mesh powders, etc.? Apparently, the average particle size of the powder should be specified? 

4) line 121: It is necessary to indicate how the compacts were prepared, what equipment was used.

5) line 126: What is meant by "ultra-pure Ar"? Apparently, the specific content of argon, the degree of its purity should be indicated.

6) line 131: Please specify the brand of microscope.

7) line 134: Missing micrometer designation (9 m).

8) lines 151, 152: It is not clear why two different types of software were used. It is necessary to detail what exactly GraphPad was used for and what ANSYS was for. 

9) line 175: The abbreviation "SHP" is not deciphered.

10) line 235: The requirement of interconnectivity of pores is completely reasonable, but the article does not contain any theoretical or experimental estimates of the transport properties of the compacts obtained. It is necessary to add at least approximate estimates.

11) line 297: Fig 7,b - mistake in a word "Frequency". 

12) line 344: Apparently, 600 micrometers, not millimeters? 

13) line 345: 20 m?

14) lines 347 - 349: The reviewer did not find any thermodynamic analysis in the manuscript; these points of conclusion are not justified. 

Author Response

1) lines 49, 50: Symbols, denoting the modification of the titanium phase, are missed;

Response: The symbols were included in the text for clarifying the text.

 2) line 53: What is the omega phase, what is its composition?

Response: The omega is a metastable phase whose composition depends on the type of system, i.e. in the Ti-Nb, its composition ranges from 16 to 45 wt.% Nb and in the Ti-Ta system it can exit in the entire composition range. This phase is fragile and has a hexagonal crystal structure with space group P6/mmm.

 3) line 113: What is -100 mesh powders, -325 mesh powders, etc.? Apparently, the average particle size of the powder should be specified?

Response: The particle size were added as clarification.

 4) line 121: It is necessary to indicate how the compacts were prepared, what equipment was used.

Response: The procedure was added in the materials and methods section.

 5) line 126: What is meant by "ultra-pure Ar"? Apparently, the specific content of argon, the degree of its purity should be indicated.

Response: The composition of Ar gas was added in the materials and methods section.

 6) line 131: Please specify the brand of microscope.

Response: The brand was specified in the materials and methods section.

 7) line 134: Missing micrometer designation (9 m).

Response: It was corrected.

 8) lines 151, 152: It is not clear why two different types of software were used. It is necessary to detail what exactly GraphPad was used for and what ANSYS was for.

Response: We agree with you. The paragraph was modified accordingly to clarigy how the two sotwares were used: the GraphPad software was used for generating the random coordinates of the pores while the ANSYS sotware was used to model the alloy/pore system. The paragraph was modified in the manuscript.

 9) line 175: The abbreviation "SHP" is not deciphered.

Response: The abbreviation was explained in the text.

 10) line 235: The requirement of interconnectivity of pores is completely reasonable, but the article does not contain any theoretical or experimental estimates of the transport properties of the compacts obtained. It is necessary to add at least approximate estimates.

Response: The interconnectivity was measured by the Cpore parameter, which was obtained by optical image analysis. Cpore values for all forams ranges between 0.3 and 0.4 which indicates that interconnectivity exists within the foams. This parameter value is 0 for null interconnectivity and 1 for maximum.

 11) line 297: Fig 7,b - mistake in a word "Frequency".

Response: The mistake was corrected.

 12) line 344: Apparently, 600 micrometers, not millimeters?

Response: The mistake was corrected.

13) line 345: 20 m?

Response: The mistake was corrected. The text should say 20 mm.

 14) lines 347 - 349: The reviewer did not find any thermodynamic analysis in the manuscript; these points of conclusion are not justified.

Response: The thermodynamic analysis was deleted from the conclusion.